# Ice-Cliff Morphometry in Identifying the Surge Phenomenon of Tidewater Glaciers (Spitsbergen, Svalbard)

**Joanna Ewa Szafraniec**

Faculty of Natural Sciences, Institute of Earth Sciences, University of Silesia in Katowice, Będzińska 60, 41-200 Sosnowiec, Poland; joanna.szafraniec@us.edu.pl; Tel.: +48-32-3689-687

**Abstract:** In this study, 110 tidewater glaciers from Spitsbergen were studied to characterize the frontal zone using morphometric indicators. In addition, their time variability was also determined based on features of the active phase of glacier surges. Landsat satellite imagery and topographic maps were used for digitalization of the ice-cliffs line. In recent years (2014–2017) all the glaciers studied can be thus classified as: stagnant (33%), retreating and deeply recessing (33%), starting to move forward/fulfilling the frontal zone (23%), and surging (11%). Indicators of the glacier frontal zone (*CfD* and *CfE*) allow to identify the beginning and the end of the active phase through changes in their values by ca. 0.05–0.06 by the year and get even bigger for large glaciers as opposed to typical interannual differences within the limits of ±0.01 to 0.02. The active phase lasted an average of 6–10 years. The presence of a "glacier buttress system" and the "pointed arch" structure of the ice-cliff seem to be an important factor regulating the tidewater glacier stability.

**Keywords:** ice-cliff morphometry; "glacier buttress system"; glacier surge; Spitsbergen; tidewater glaciers

## 1. Introduction

Spitsbergen glaciers are in rapid recession. This is observed in both land-based glaciers [1–5] and those terminating at sea [6–8] and is regarded as a manifestation of the Arctic amplification effect, whereby changes in the net radiation balance tend to produce a larger increase in temperature near the pole than the planetary average [9,10]. This process contributes to the expansion of low-albedo, ice-free sea surfaces with positive feedback (e.g., [11–13]).

In Spitsbergen (Svalbard Archipelago), surging glaciers represent a significant proportion of the glacier cover [6,14–22]. They operate within the so-called "climate envelope" [23], i.e., in optimal thermal and humidity conditions. Their behavior is mainly driven through snow mass supply in the glacier reservoir area (e.g., [24,25]) and glacier enthalpy disorder, mainly at the ice-cliff area [22,26]. The surging tidewater glaciers behave differently than the land-based glaciers. On land, the increased stress mainly propagates through the glacier surface from the accumulation area to the ablation zone where the maximum ice flow velocities are also measured. This sometimes manifests as an ice bulge [19,27,28]. For tidewater glaciers, on the other hand, the properties of the active phase are often more obvious at the glacier ice-cliff. This phenomenon tends to be seen as an expanding crevassed zone [22], as a result of the steep slope on the glacier surface propagating due to ablation of its lowermost part—the frontal zone. These features have been observed in Aavatsmarkbreen, for example. Other glaciers also show a characteristic lowering of the frontal zone due to ablation, and consequently intensive calving and retreat into deeper water from the pinning point. This is, for example, the case of Wahlenbergbreen, Mendeleevbreen, or Paierlbreenn [6]. Meltwater and precipitation entering the glacier also enhances crevasse propagation towards the top of the glacier [22,29,30]. The presence of crevasses in the frontal

zone is important for determining tidewater glacier dynamics. In particular, crevasse formation increases the susceptibility of the glacier to mechanical ablation by calving [6,7,20,31,32], facilitates the distribution of meltwater through en- and subglacial drainage systems [22,33–35], particularly at low inclinations, crevasse allows penetration of direct radiation into deeper parts (also through multiple reflection of radiation from the crevasse ice walls [36]) and facilitates ice-atmosphere and ice-ground/sea thermal conductivity. The length of the ice-cliff [20] also influences thermal conductivity processes at the common ice–water–atmosphere boundary. As such, the ice-cliff wall also represents an additional heat and energy exchange zone. The glacier behavior is also influenced by the depth of the fjord into which the glacier flows [37–39]. Thus, glaciers terminating in deeper parts of fjords are more susceptible to form a dense network of crevasses, further supplying heat to the underwater sections of the ice-cliff. This effect is particularly pronounced in areas where warmer West Spitsbergen Current waters penetrate deeply into the fjord [40].

Regardless of the mechanisms determining the surging behaviour, a common sign of the active phase from all surging glaciers is the quick and constant change in geometry of the glacier frontal zone. If the ice-cliff bent towards the sea (convex), it is indicative of an advancing phase, whilst a bending towards the land (concave) is a sign of glacier retreat through intensive calving. This geometry is indeed representative of its dynamic state.

The purpose of this study is to analyze Spitsbergen tidewater glaciers using selected morphometric parameters of the frontal zone. This study combines data from 110 tidewater glaciers, including surging glaciers. Here, data from 1936 to 2017, both from recessive and advancing fronts is analyzed. In particular, the study focuses on annual geometrical changes of glaciers surging in the 1985–2017 period. Frontal morphometry was thus assessed as potential indicator to determine the initiation and termination of the active phase of the glacier surge. These data was used to forecast the initiation of active phases in 2017 and later, based on frontal morphology data from the 2014–2017 period. The results suggest that the "glacier buttresses system" of the frontal zone anchored on land plays a potential role on ice-cliff stabilization.

## 2. Materials and Methods

Landsat 7 Enhanced Thematic Mapper Plus (ETM+) satellite images with 30 m × 30 m resolution (543 bands) were collected from the summer seasons in year 2000 supplemented with a few in 2002. Landsat 8 Operational Land Imager/Thermal InfraRed Sensor (OLI/TIRS) imagery (654 bands) were collected for the 2014, 2017, 2018, and 2019 seasons (Table S1: Emblems of topographic maps and IDs and acquisition dates of Landsat satellite images used in the research). These images were downloaded using the EarthExplorer browser [41]. They were published by the United States Geological Survey (USGS) and the National Aeronautics and Space Administration (NASA) in the public domain. The selected data cover the ablation season. Topographic maps at 1:100,000 scale published by Norwegian Polar Institute (old and new editions [42]) were used to define the ice-cliff edges for the years 1936, for the 1960s and 1970s, and partly for the 1990s. The maps from the older edition were georeferenced to the UTM 33 coordinate system using the ED50 ellipsoid (EPSG:23033) and then transformed into ETRS89 (EPSG:25833). Older Landsat 5 Thematic Mapper (TM) (543 bands) images from the period 1985–1998 and data described earlier were used to track yearly changes in the morphometry of 15 selected glaciers (1985–2017) (Table S2. IDs and acquisition dates of Landsat satellite images for selected tidewater glaciers in Spitsbergen in the period 1985–2019). For a few cases Terra ASTER © NASA satellite scenes were used. Collected satellite images were also used to digitize the surface area of glaciers on available imagery in the year (max. 5 years) before the occurrence of the active phase of the glacier surge (see Table S2—data pointed by red color of font). The vector layers (polygons) were used to calculate the basic morphometric parameters of a given glacier in order to examine the relationship between them and morphometry of the frontal zone. The average glacier slope is determined as the ratio between the elevation difference (maximum and minimum altitudes of the glacier found using the TopoSvalbard portal [43]) and the length of the glacier measured along the

central line. The compactness coefficient was calculated as the ratio of the glacier perimeter to the circle perimeter of area equal to the glacier area.

The data were analyzed using QGIS software (different versions). The ice-cliff line was delineated (vector layers) manually in a 1:20,000 scale. The composition of 543 (654) spectral bands clearly highlights the glacier ice area (blue), the non-glaciated area on land (brown, red) and sea water area (black, dark navy blue). For maps, the glacier front delineated in the original study was used as the cliff line.

The length of the ice-cliff line may vary between seasons, particularly depending on whether the image was taken at the beginning or the end of the ablation season. Uniform quality images from the whole of Spitsbergen are not available from the same day, due to variable cloud cover, different glacier exposure (shading), or other variables such as the presence of pack ice. The glacier range defined in the maps used were also based on aerial photographs and satellite images taken on different days for the same reasons.

This study assumes that the glacier frontal zone reacts clearest and fastest to changes in glacier dynamics than any other glacier sections. The following definition of the frontal zone (Figure 1) was used here as standard: the "frontal zone Ag" of tidewater glaciers is defined as the section contained within the *Ac* circle with a diameter *Dc* equal to the distance between the parts of the glacier ice-cliff *Lc* anchored on land.

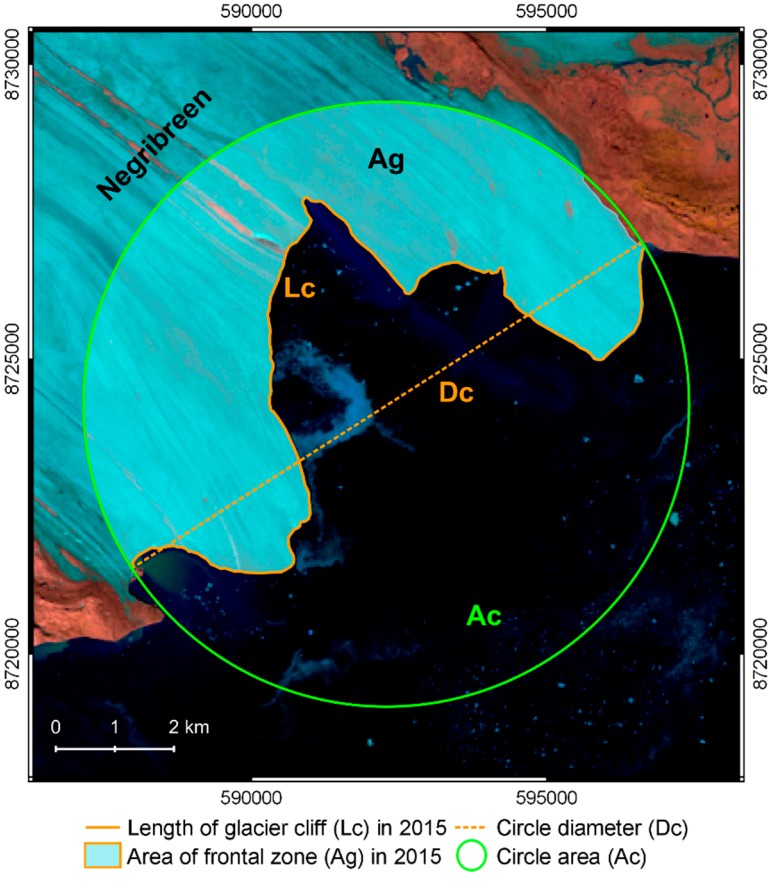

**Figure 1.** Frontal zone Ag of Spitsbergen tidewater glaciers on the example of Nigerbreen (based on the 654 spectral bands composition of the Landsat 8 satellite imagery ID: LC82150032015238LGN01, 26.08.2015, USGS/NASA Landsat [41].

Following the definition of the glacier frontal zone, the morphometric indicators—"glacier front dynamics indicator *CfD*" and "ice-cliff balance indicator *CfE*"—were defined. Thus, *CfD* represents the ratio of glacier frontal zone area *Ag* to circle area *Ac*:

$$CfD = \frac{Ag}{Ac}. \tag{1}$$

A *CfD* close to 1 or higher represents a convex frontal zone typical of advancing glaciers (Figure 2a). The highest absolute value of *CfD* = 1.11 was found for Negribreen in 1969. A value of *CfD* above 1 (extreme cases) usually means that the frontal zone has spilled out and moved far out into the sea, beyond the outlet of the valley that limits it, and in consequence beyond the perimeter of the theoretical circle *Ac*. A *CfD* of approximately 0.5 represents a semi-circular frontal zone with a cliff line closely following the diameter of the circle and usually presenting very little variability (Figure 2b). Finally, a *CfD* close to or below 0 is indicative of a clearly concave glacier front in deep recession, where the inflexion point(s) of the ice-cliff line is often at a long distance from the line between the anchor points (Figure 2c). A negative value appears when the greater part of the frontal zone is outside the circle area *Ac*. Then this "minus" area needs to be subtracted from the value of the frontal zone area lying inside the circle (usually lateral part anchored on land). The lowest *CfD* = −1.15 was found for Samarinbreen in 1961.

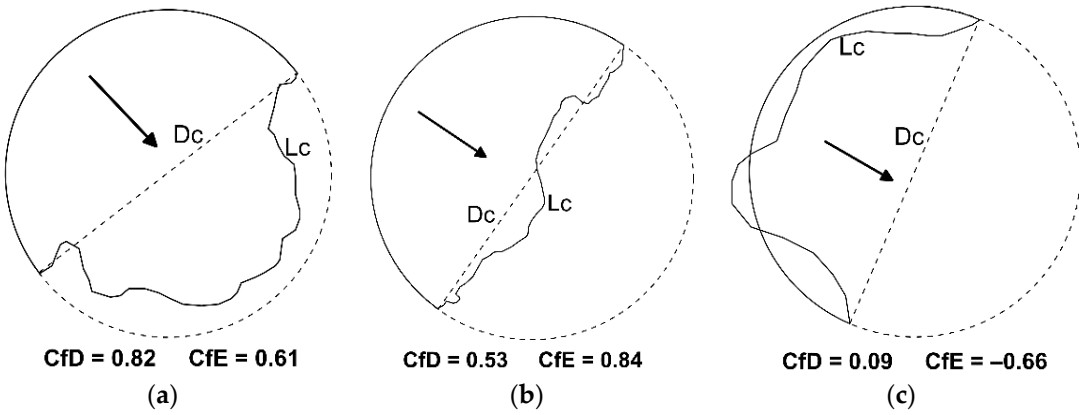

CfD = 0.82    CfE = 0.61        CfD = 0.53    CfE = 0.84        CfD = 0.09    CfE = −0.66

(**a**)          (**b**)          (**c**)

**Figure 2.** Examples of the shape of the frontal zone expressed by *CfD* and *CfE* indicators depending on the dynamic state of a tidewater glacier: (**a**) Advancing glacier; (**b**) Glacier in the momentary equilibrium state; and (**c**) Glacier undergoing recession. An arrow points the glacier flow direction.

The ice-cliff balance indicator, *CfE*, is the ratio of the circle diameter length *Dc* to the ice-cliff length *Lc*:

$$CfE = \frac{Dc}{Lc}. \tag{2}$$

The absolute value (a.v., modulus), used to compared among different glaciers, represents the degree of curving of the ice-cliff. Thus, an absolute of 1 indicates a straight line; that is, that the length of the ice-cliff *Lc* is equal to the circle diameter *Dc*. This can be achieved by any glacier during the evolution between the concave and convex shape of its frontal zone (and vice versa). Ice-cliff bending level increases as the absolute values approach 0, with the number of inflexion points increasing concomitantly. The greatest curvature of the ice-cliff was found for Sefströmbreen in 1990 (|*CfE*| = 0.21). The symbols + and − are used to distinguish between convex (+) and concave (−) glaciers. A negative sign is when most of the measured ice-cliff length *Lc* is located between the circle diameter *Dc* and the part of the frontal zone closer to the glacier body. In turn, when most of the cliff line exceeds the circle diameter *Dc* towards the sea, the sign for *CfE* is positive (cf. Figure 2).

## 3. Results

### 3.1. Geometric Parameters of the Glacier Ice-Cliffs in Spitsbergen (1936–2017)

Ice geometry data from Spitsbergen glaciers (Table 1) were compiled for the year 1936 (28 glaciers; no data available from the NW and N parts of Spitsbergen), the period 1961–1966 (35 glaciers; no data available from the N and NE parts of Spitsbergen), 1969–1977 (17 glaciers; no data available from S Spitsbergen), 1990 (73 glaciers), 2000, 2014, and 2017 (110 glaciers each year). The different periods include different combinations of glacier systems (their complexity). For example, Strongbreen was configured as one glacier system in 1936 while it is currently shaped as several separate land-terminated (e.g., Karlbreen) and tidewater glaciers (e.g., the Kvalbreen–Indrebøbreen system, Moršnevbreen, the Sokkelbreen–Naglebren–Nuddbreen system ⇒ Strongbreen, the Vindeggbreen–Persejbreen system), as a result of recession.

Due to the asymmetrical distribution of morphometric parameters of the glaciers in terms of their area and their frontal zones, the calculation of average values for the glaciers population from each period took into account the median and interquartile range (IQR) values. Glaciers between 19.5 $km^2$ and 110.4 $km^2$ analyzed here (within the IQR; based on data by Błaszczyk et al. [20]) can be considered as typical in terms of surface area.

**Table 1.** Median values of morphometric parameters of glacier frontal zones of Spitsbergen tidewater glaciers (the interquartile range IQR is given in brackets).

| Year | Lc [km] | Dc [km] | Ag [$km^2$] | Ac [$km^2$] | CfD | CfE (a.v.) |
|---|---|---|---|---|---|---|
| 1936 | 4.24 | 3.35 | 3.62 | 8.82 | 0.40 | 0.79 |
| (28 glaciers) | (3.43–5.74) | (2.31–4.10) | (1.01–6.24) | (4.18–13.26) | (0.35–0.45) | (0.71–0.87) |
| 1960–1966 | 2.47 | 1.85 | 0.91 | 2.67 | 0.37 | 0.75 |
| (35) | (1.93–4.71) | (1.43–3.05) | (0.43–1.92) | (1.60–7.24) | (0.30–0.43) | (0.63–0.81) |
| 1969–1977 | 5.58 | 3.03 | 2.71 | 7.21 | 0.38 | 0.61 |
| (17) | (4.38–6.61) | (2.39–3.43) | (−0.87–4.57) | (4.47–9.19) | (−0.20–0.48) | (0.40–0.81) |
| 1990 | 3.16 | 2.24 | 0.93 | 3.93 | 0.30 | 0.60 |
| (73) | (2.28–5.09) | (1.44–3.12) | (0.11–2.46) | (1.62–7.69) | (0.16–0.40) | (0.30–0.80) |
| 2000 | 2.71 | 1.71 | 0.74 | 2.30 | 0.39 | 0.79 |
| (110) | (1.17–3.73) | (1.00–2.66) | (0.20–1.92) | (0.79–5.55) | (0.27–0.47) | (0.66–0.87) |
| 2014 | 2.78 | 1.72 | 0.59 | 2.34 | 0.30 | 0.71 |
| (110) | (1.39–4.38) | (1.00–2.89) | (0.19–1.89) | (0.78–6.59) | (0.18–0.40) | (0.57–0.79) |
| 2017 | 2.61 | 1.86 | 0.69 | 2.70 | 0.29 | 0.70 |
| (110) | (1.46–4.23) | (1.04–3.01) | (0.08–1.95) | (0.85–7.12) | (0.13–0.44) | (0.57–0.80) |

For the following parameters—*Lc*, *Dc*, Ag, and Ac—the results show that the highest median appeared in 1936. The values subsequently decreased until the year 2000. Thus, the median ice-cliff length (*Lc*) decrease by 1.6-fold, the median valley width at the glacier front anchoring points (*Dc*) showed a 2-fold decrease, the frontal zone area (Ag) decreased by 4.9-fold, and the median value of area of the circle surrounding the glacier frontal zone (Ac) decreased by 3.8-fold. However, the period 1969–1977, when the values were slightly higher, was an exception. In addition, an increase in *Lc* was observed between 2000 and 2014 followed by a 1.1-fold decrease. Ag, on the other hand, showed a 1.3-fold decrease first followed by a 1.2-fold increase measured in 2017. The *CfD* and *CfE* indicators also show a general downward trend, suggesting that ice-cliffs are presenting an increasingly recessive nature. The exception was the year 2000, when these indicators increased.

If the interaction between *CfD* and *CfE* for each glacier is considered, then most of the glaciers were in a state of recession throughout all the periods discussed here (negative *CfE* values and *CfD* values below 0.5) (Figure 3). However, there is always a group of advancing glaciers (or shortly after the active phase of the glacier surge), representing 10.5% (1990) to 18.2% (2000) of the population of

tidewater glaciers. These glaciers are characterized by a positive *CfE* value and a *CfD* value often above 0.5.

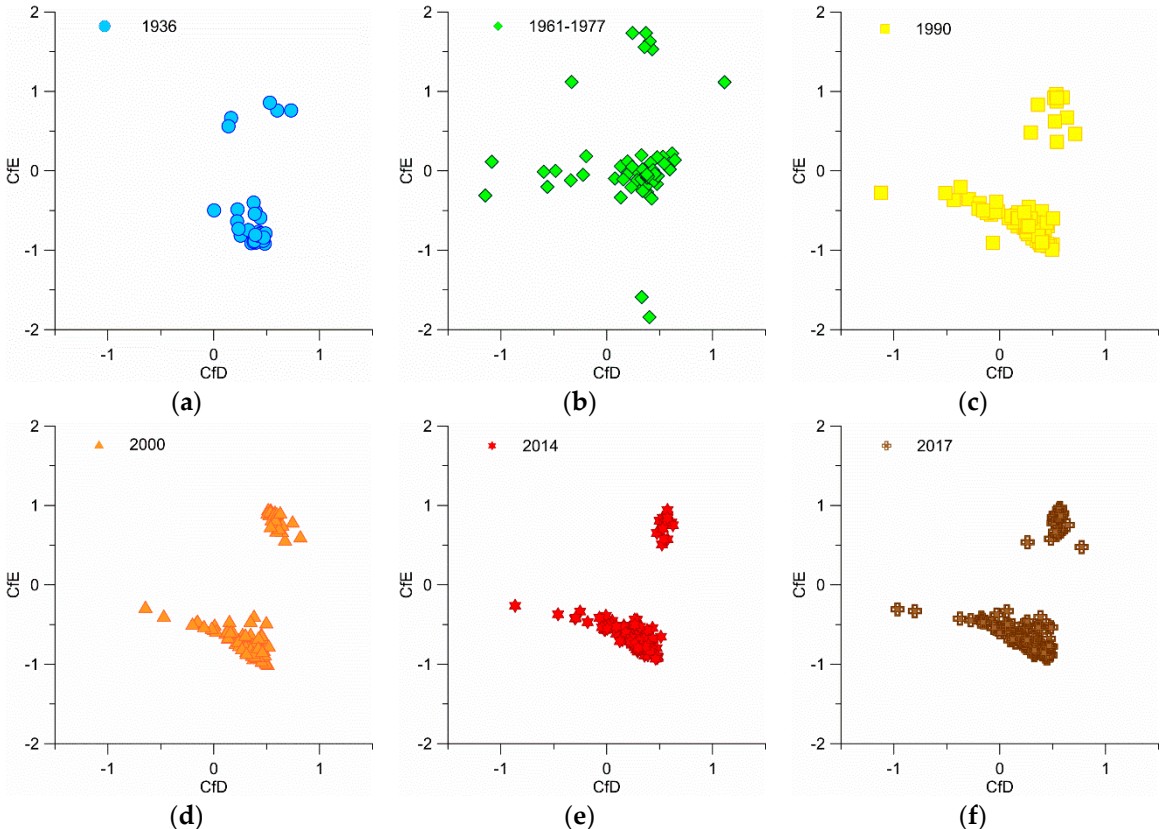

**Figure 3.** Relationships between the glacier front dynamics indicator *CfD* and the ice-cliff balance indicator *CfE* for Spitsbergen tidewater glaciers in selected years of the 1936–2017 period: (**a**) 1936; (**b**) 1961–1977; (**c**) 1990; (**d**) 2000; (**e**) 2014; and (**f**) 2017.

### 3.2. Frontal Zone Morphometry of the Retreating Glaciers

The values of morphometric indicators of the glacier frontal zones during recession phase were plotted against the mean annual air temperature (MAAT) measured at the Svalbard Airport for each of the different periods considered (Figure 4). Glaciers showed a less intense recession following the minimum temperatures recorded at the beginning of the 20th century, presenting the highest median *CfD* and *CfE* (specifically, *CfD* = 0.39 and *CfE* = 0.81 a.v. in 1936). In contrast, the same glaciers showed their strongest recession episode over 30–40 years later, where the median values for both indicators dropped to *CfD* = 0.24 and *CfE* = 0.63 a.v., (median from 1969–1977). Until the end of the 20th century, the median values increased reaching *CfD* = 0.36 and *CfE* = 0.78 a.v. in 2000. The recession process further intensified over the first 17 years of the 21st century, accompanied by a decrease in *CfD* and *CfE*, down to 0.26 and 0.69 a.v., respectively, in 2017. Both values were close to the levels observed in the 1990s, which was followed by the largest number of surging tidewater glaciers in Spitsbergen.

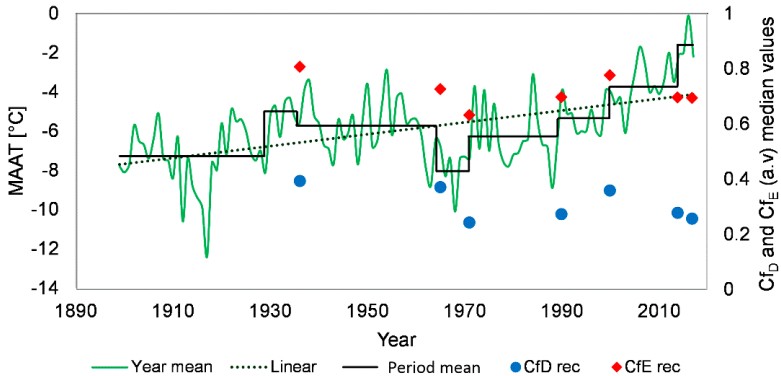

**Figure 4.** Median values of the *CfD* and *CfE* indicators for the selected years of the 1936–2017 period under the background of the mean annual air temperature at the Svalbard Airport (temperature values based on data made available by the Norwegian Meteorological Institute [44].

In relation to ice-cliff exposure in 1990, south-east and north-west facing glaciers showed the strongest recessive nature (Figure 5). These glaciers showed a median *CfD* of 0.11 and a relatively low *CfE* (a.v.) compared to the other glacier groups.

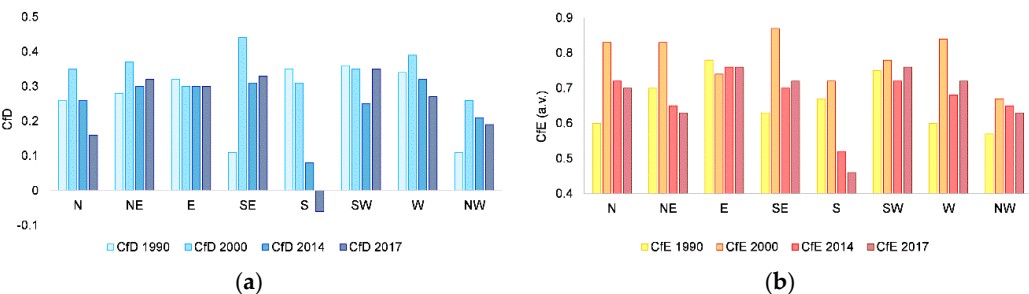

**Figure 5.** Median values of *CfD* (**a**) and *CfE* (a.v.) (**b**) indicators relative to the exposition of the Spitsbergen tidewater glaciers during the recession phase in years: 1990, 2000, 2014, and 2017.

In the year 2000, a clear growing trend was observed for the dynamically active of most glacier fronts, apart from those from the southern sector. This was most noticeable for ice-cliffs facing the south-east sector, with *CfD* values reaching over 0.44. The ice-cliff lengths in relation to the circle diameter also shortened, presenting median *CfE* (a.v.) of approximately 0.80.

More recessive stage was also observed for the 2014–2017 period, apparent as a clearly visible melting of the most dynamic part of the glacier frontal zones, particularly obvious for ice-cliffs facing to the southern sector. At this stage, ice-cliffs withdraw considerably with the *CfD* reaching its lowest recorded value at −0.06. This curving is also reflected in the lowest median *CfE* reaching 0.46 (a.v.). A significant decrease in *CfD* and *CfE* was also observed from north-facing glaciers, and to a lesser extent, for west- and north west-facing fronts. The fluctuation in *CfD* and *CfE* was smaller for east- and south west-facing glaciers.

*3.3. Frontal Zone Morphometry of the Advancing Glaciers*

Over the years 1990–2017, advancing ice-cliffs were found in glaciers from all geographical aspects, and roughly around the same number of cases (up to 5% difference). This pattern was particularly obvious for all north-, east-, or south west-facing tidewater glaciers from Spitsbergen, with fewer cases found from north-east (8% difference), south, and west. However, the proportion of cases changed over this period. Thus, in 1990–2000 the most active glacier fronts were found particularly in the N–S axis (e.g., Petermannbreen, Monacobreen, Waggonwaybreen, Paierlbreen, and Storbreen) and with north-east exposure (e.g., Davisbreen, and Inglefieldbreen-Nordsysselbreen). In turn, in 2014–2017, active ice-cliffs dominate in the E–W axis

(e.g., Vindeggbreen-Persejbreen, Sjettebreen, Arnesenbreen, and Strongbreen) and from south west-facing glaciers (e.g., Blomstrandbreen, Kollerbreen, and Marstrandbreen).

Annual variations in the frontal zone geometrical indicators were analyzed for 15 glaciers. These glaciers experienced an active phase of the glacier surge at some point during 1985–2017 (based on: Murray et al. [45]; Błaszczyk et al. [20]; Sund et al. [19]), or/and the advance could be directly observed from data analyzed in this study (Figure 6, Table 2). As the glaciers advance, their *CfD* values changed from 0.33 (0.18 to 0.40 of IQR) during the quiescent phase to 0.61 (0.54 to 0.71 of IQR) during the active phase. The median *CfE* changed from −0.72 (−0.81 to −0.56 of IQR) during the quiescent phase to 0.78 (0.69 to 0.88 of IQR) during the active phase of the glacier surge.

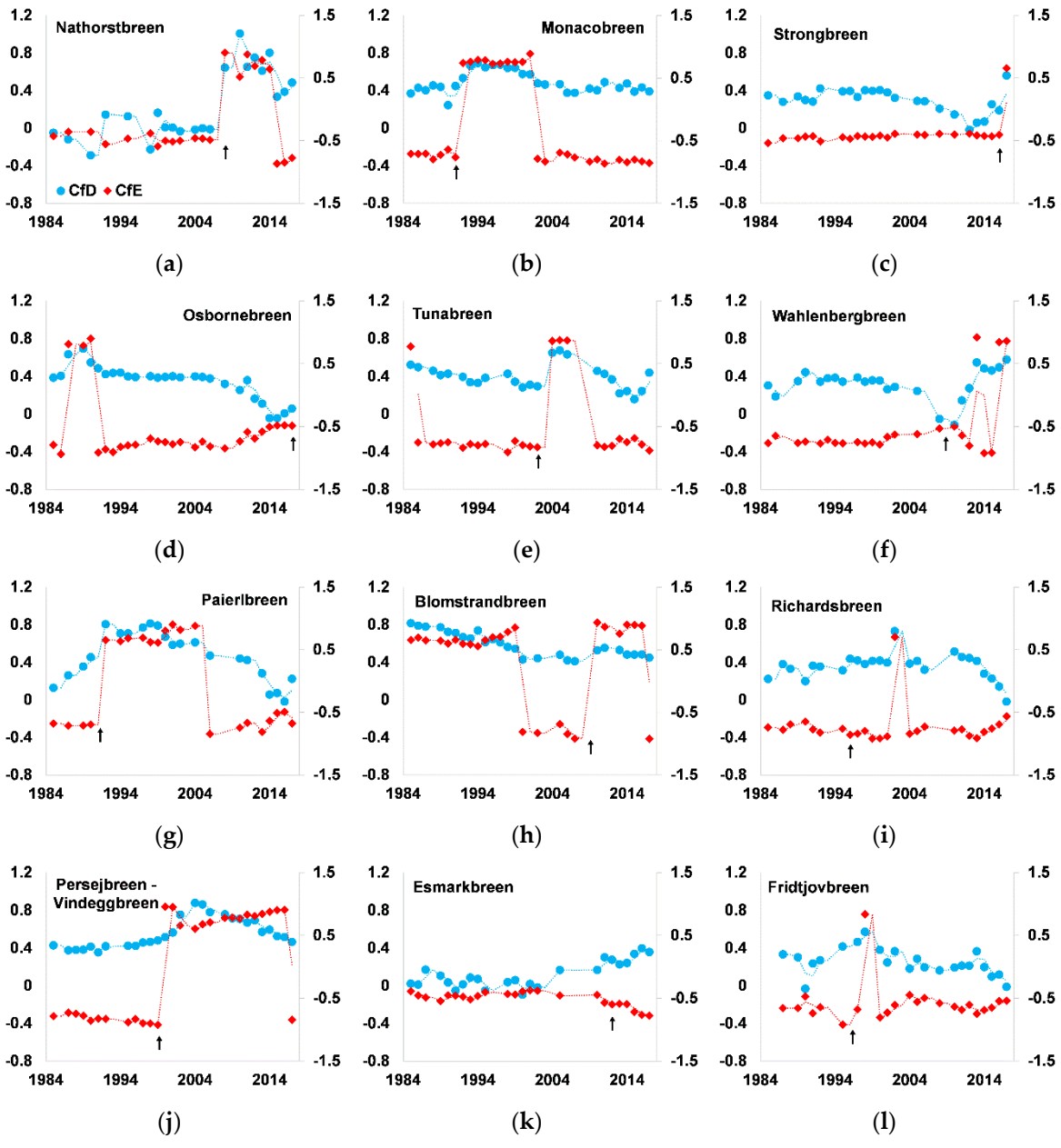

**Figure 6.** *Cont.*

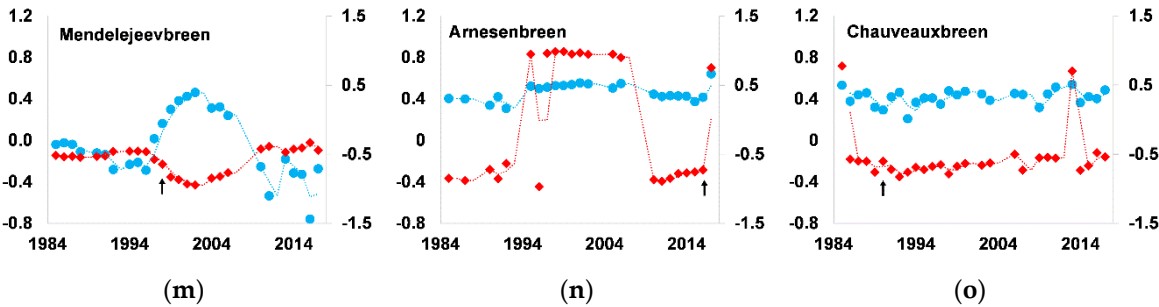

**(m)**          **(n)**          **(o)**

**Figure 6.** *CfD* (left Y axis; blue circles) and *CfE* (right Y axis; red rhombs) values for selected tidewater Spitsbergen glaciers, which experienced an episode of the active phase of the glacier surge in 1985–2017 (X axis–years): (**a**) the Nathorstbreen–Zawadzkibreen system; (**b**) Monacobreen; (**c**) the Strongbreen–Moršnevbreen system; (**d**) the Osbornebreen–Vintervegen system; (**e**) Tunabreen; (**f**) Wahlenbergbreen; (**g**) Paierlbreen; (**h**) Blomstrandbreen; (**i**) Richardsbreen; (**j**) the Persejbreen–Vindeggbreen system; (**k**) Esmarkbreen; (**l**) Fridtjovbreen; (**m**) Mendelejeevbreen; (**n**) Arnesenbreen; and (**o**) Chauveauxbreen. An arrow indicates the moment of the glacier movement forward.

**Table 2.** The beginning and the end of the active phase of the glacier surge for selected Spitsbergen tidewater glaciers based on publications and morphometric indicators of the glacier frontal zone.

| Surging Glaciers | Source of Data (Published) | According to Published Data | | According to the Ice-Cliff Morphometry | |
|---|---|---|---|---|---|
| | | Start of the Active Phase | End (and Duration—in Brackets) of the Active Phase | Start of the Active Phase | End (and Duration—in Brackets) of the Active Phase |
| Nathorstbreen system | Sund et al. [46] | 2003–stage 1 2008–stage 2 | 2013 (?) (~5–10 years) | 2006–2008 | 2014–2015 (6–9 years) |
| Monacobreen | Murray et al. [45] Mansell et al. [27] | 1990–1992 1993–1995 | 1997 (2–7 years) | 1991–1992 | 2001–2002 (9–10 years) |
| Osbornebreen | Dowdeswell et al. [14]; Rolstad et al. [47] | 1986–1987 | ? | 1986–1987 | 1990–1991 (3–5 years) |
| Tunabreen | Flink et al. [48]; Sevestre et al. [49] | 2002–2003 2003 | 2004–2005 2005 (1–3 years) | 2002–2004 | 2006–2010 (2–8 years) |
| Wahlenbergbreen | Sevestre at al. [22] | 2009 | – | 2012–2013 | – |
| Paierlbreen | Błaszczyk et al. [6] | 1993 (?) | 1999 (?) (6 years?) | 1990–1992 | 2004–2006 (12–16 years) |
| Blomstrandbreen | Mansell et al. [27]; Burton et al. [50] | 2007 2009 | 2010 (?) 2013 (1–6 years) | 2007–2010 | 2016–2017 (6–10 years) |
| Persejbreen | Dowdeswell and Benham [51] | 2000–2001 | ? | 1999–2000 | 2015–2016 (5–17 years) |
| Fridtjovbreen | Murray et al. [45]; Murray et al. [52]; Lønne [53] | 1994–1995 and re-advance in 1998–1999 | 1997 (2–3 years or 3–5 years) | 1995–1996 | 1998–1999 (2–4 years) |
| Mendeleevbreen | Błaszczyk et al. [6] | between 1995–2002 | between 2002–2010 (up to 15 years?) | 1996–1997 | 2006–2010 (9–14 years) |

The comparison presented in Table 2 allows assessing how the timing of the initiation and end of the active phase can be determined based on *CfD* and *CfE*. This relation differed slightly for the published data: by ±0 to 4 years for the beginning and ±1 to 6 years for the end of the active phase. The average duration of the active phase based on data published for 10 glaciers was approximately 3–10 years, while the *CfD* and *CfE* indicators showed an average active phase of 6–10 years. Some of the published results, however, appeared in print during the active phase and the glacier could have continued surging afterwards.

The increase in *CfD* and *CfE* was nearly concomitant to the glacier movement for larger glaciers (e.g., Nathorstbreen, Monacobreen, and Paierlbreen). However, a delay of few years was also observed (e.g., Mendeleevbreen and Wahlenbergbreen). A correlation analysis between the values of both indicators also show a delay of up to five years before the indicator values jump (Table 3). The largest significant correlation of indicators at the significance level of 0.05 took place one year and five years before the active phase. In all cases, the active phase ended with an obvious decrease in *CfD* and *CfE*; thus, entering a quiescent phase characterized by small interannual changes in these values.

**Table 3.** The relationship between *CfD* and *CfE* up to 5 years before the active phase of the glacier surge expressed as a correlation coefficient.

| Year Before Surge | 1 | 2 | 3 | 4 | 5 |
|---|---|---|---|---|---|
| Coeff. of corr. | −0.84 | −0.73 | −0.76 | −0.77 * | −0.80 |

\* Not statistically significant.

Preliminary analysis indicates that the length of the ice-cliff just before the active phase of the glacier surge is well correlated with the glacier surface slope and the glacier compactness. The gentler the slope of the glacier surface and more complex the glacier in terms of its geometry, the longer its ice-cliff. This is because the glaciers that are most complex in shape are usually large valley glacier systems with numerous tributaries. These studies will be continued.

*3.4. Application of CfD and CfE Indicators in the Classification of Spitsbergen Tidewater Glaciers in Terms of Dynamics*

Interannual changes in indicator values were presented to describe the dynamic state of five Spitsbergen tidewater glaciers with the most complete data sets for the period 1985–2017: Monacobreen (to 2019; N part of Spitsbergen), the Vindeggbreen–Persejbreen system (E part of the island), the Osbornebreen–Vintervegen system and Blomstrandbreen (both in W part of Spitsbergen), and Paierlbreen (S part of the island) (Figure 7; see Table S2). The data indicates that most of the time the ice-cliffs are characterized by slight interannual changes in indicator values. The situation changes significantly with the beginning of the active phase of the glacier surge when the increase in values appears as a clear deviation in the I quadrant of the graph (advance), and this is also recorded in Figure 6. A similar clear deviation occurs in the III quadrant during the transition of the active phase to the quiescent phase. Then in Figure 6 the *CfE* values go negative (retreat).

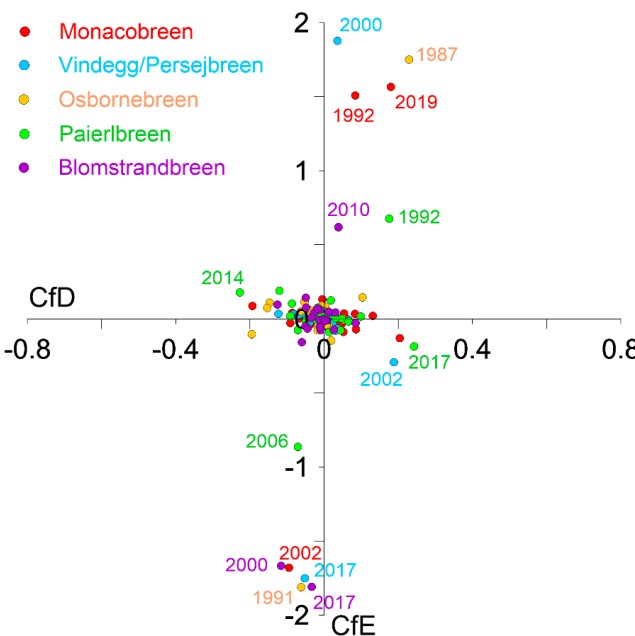

**Figure 7.** Interannual changes in the values of *CfD* and *CfE* indicators for five selected tidewaters glaciers surging between 1985 and 2017.

The analysis of the interannual variation in the *CfD* and *CfE* indicators allows describing the following dynamic models for tidewater surging glaciers in Spitsbergen:

1. At the beginning of "the active phase of the glacier surge" both *CfD* and *CfE* indicators increase rapidly by 3–5 times during approximately 1–5 years. For larger glaciers, the rate of increase can reach a maximum of a 10× increase (I quadrant in Figures 7 and 8). In this phase, the frontal zone protrudes (convex shape) and the glacier moves forward. Then the ice-cliff position is marked by slight fluctuations or "stagnates", and numerous inflexion points associated with intensive calving appear in the cliff line.

2. After the active phase, the glacier first loses its frontal zone convexity, mainly as a result of intensive calving, and it is subject to "recession" (including the quiescent phase of the glacier surge). At this point, *CfD* and *CfE* values decrease interannually to an average of 0.05–0.06 (for *CfE* the sign changes to negative; III quadrant).

3. Subsequently, the glacier can enter a "deep recession phase", when its frontal zone is strongly concave, especially for the largest glaciers (IV quadrant). "Glacier buttresses" are also observed, i.e., parts of the glacier front anchored on land, from which the ice-cliff bends into an arch. During the quiescent phase, *CfD* and *CfE* values change very little interannually and are ca. 0.015–0.025. The value of *CfD* decreases the most (it can even be negative), with the *CfE* value slowly increasing from −1 to approximately −0.5.

4. The glacier begins to lose its maximum ice-cliff concavity at the end of the recession with "the frontal zone filling and the slow forward movement" beginning (II quadrant). At this point, the *CfD* value slowly increases and *CfE* value slowly decreases (from −0.5 to −1) until the protrusion of the glacier front and the value changes to plus sign.

The presented model was adopted for Monacobreen for the 1985–2019 period (Table 4). Based on the indicators, the beginning of the active phase (I) of the glacier surge was determined in 1991–1992. For the first 2 years, the frontal zone was getting convex and the glacier advanced about 1100 m in western part to about 1500 m in eastern part of the cliff line compared to 1991. Then it entered a period of stagnation/minor fluctuations in the position of its ice-cliff, rising slightly in the western part and about 700 m in the eastern part of the cliff. At the end of the active phase (2000–2001), retreat began with the still convex frontal zone. The active phase lasted until 2001–2002 (10 years). After this

period, the glacier entered the quiescent phase of the glacier surge (negative *CfE*). Slow withdrawal of the ice-cliff position was accompanied by ablation by calving and melting (concave shape). By 2007, the glacier withdrew from about 700 m in the eastern part of the ice-cliff to about 800–900 m in the western part. The appearance of inflexion points (2005) indicates probably an important role of the outflow of subglacial waters. After 2008, the quiescent phase was marked by some recovery in glacier behaviour, reflected in changes in the indicators values from year to year up to 2018. The glacier retreated by 500–600 m. The quiescent phase lasted 17 years. In 2018–2019, a new active phase (II) of the glacier surge began (positive *CfE*), as before, from the bending of the ice-cliff line towards the sea and an advance of about 800–900 m on the central line of the glacier.

**Table 4.** Characteristics of Monacobreen dynamics based on *CfD* and *CfE* indicators and their interannual changes in 1985–2019.

| Year | The Ice-Cliff Shape | Indicators | | Interannual Changes in Indicators Value | | | Dynamics |
| --- | --- | --- | --- | --- | --- | --- | --- |
| | | CfD | CfE | Ch. of CfD | Ch. of CfE | | |
| 1985 |  | 0.367 | −0.713 | ? | ? | ? | ? |
| 1986 |  | 0.428 | −0.711 | 0.061 | 0.002 | | |
| 1987 |  | 0.402 | −0.711 | −0.026 | 0.000 | | Frontal zone filling/stagnation |
| 1988 |  | 0.454 | −0.799 | 0.052 | −0.088 | | |
| 1989 |  | 0.437 | −0.729 | −0.017 | 0.070 | | Deep recession stage—strongly concave ice-cliff shape |
| 1990 |  | 0.244 | −0.641 | −0.193 | 0.088 | | |
| 1991 |  | 0.448 | −0.770 | 0.204 | −0.129 | | Entering the active phase (I) of the glacier phase—an advance, convex shape of the ice-cliff |
| 1992 |  | 0.532 | 0.738 | 0.084 | 1.508 | | |
| 1993 |  | 0.663 | 0.758 | 0.131 | 0.020 | | |
| 1994 |  | 0.691 | 0.789 | 0.028 | 0.031 | | |
| 1995 |  | 0.646 | 0.783 | −0.045 | −0.006 | | |
| 1996 |  | 0.670 | 0.724 | 0.024 | −0.059 | | Stagnation—convex ice-cliff, many inflexion points |
| 1997 |  | 0.675 | 0.732 | 0.005 | 0.008 | | |
| 1998 |  | 0.640 | 0.759 | −0.035 | 0.027 | | |
| 1999 |  | 0.641 | 0.750 | 0.001 | −0.009 | | |

Table 4. *Cont.*

| Year | The Ice-Cliff Shape | Indicators | | Interannual Changes in Indicators Value | | | Dynamics |
|------|---------------------|------------|------|------------------------------------------|------|------|----------|
| | | CfD | CfE | Ch. of CfD | Ch. of CfE | | |
| 2000 | | 0.575 | 0.753 | −0.066 | 0.003 | | Deep recession by calving and melting |
| 2001 | | 0.570 | 0.885 | −0.005 | 0.132 | | |
| 2002 | | 0.475 | −0.794 | −0.095 | −1.679 | | |
| 2003 | | 0.463 | −0.837 | −0.012 | −0.043 | | |
| 2004 | − | − | − | − | − | − | |
| 2005 | | 0.466 | −0.695 | ? | ? | ? | |
| 2006 | | 0.375 | −0.722 | −0.091 | −0.027 | | Entering the quiescent phase of the glacier stage—retreat, few inflexion points on the ice-cliff line |
| 2007 | | 0.375 | −0.770 | 0.000 | −0.048 | | |
| 2008 | − | − | − | − | − | − | |
| 2009 | | 0.420 | −0.839 | ? | ? | ? | |
| 2010 | | 0.402 | −0.801 | −0.018 | 0.038 | | |
| 2011 | | 0.488 | −0.870 | 0.086 | −0.069 | | |
| 2012 | − | − | − | − | − | − | |
| 2013 | | 0.427 | −0.809 | ? | ? | ? | |
| 2014 | | 0.474 | −0.845 | 0.047 | −0.036 | | Small interannual fluctuations and stagnation—filling the frontal zone balanced by the ice-cliff retreating |
| 2015 | | 0.388 | −0.804 | −0.086 | 0.041 | | |
| 2016 | | 0.431 | −0.835 | 0.043 | −0.031 | | |
| 2017 | | 0.390 | −0.858 | −0.041 | −0.023 | | |
| 2018 | | 0.473 | −0.823 | 0.083 | 0.035 | | Entering the active phase (II) of the glacier phase—an advance, convex shape of the ice-cliff |
| 2019 | | 0.653 | 0.743 | 0.180 | 1.566 | | |

Explanation: "The ice-cliff shape" column: solid line—*Lc*; dashed line—*Dc*; the "Indicators" column: pink colour—active phase; blue colour—quiescent phase; colors in the "Dynamics" column—see Figure 8; and stagnation—values centered around 0 of interannual changes within the IQR: from −0.037 to 0.054 for *CfD* and from −0.038 to 0.037 for *CfE*).

Interannual variations in *CfD* and *CfE* estimated on the basis of the data from 2014 and 2017 were also analyzed including all glaciers (Figure 8) to determine which of them may experience an active phase of glacier surge in 2017 and later according to the model proposed. In addition, a class

represented stagnant glaciers has been specified based on the IQR of interannual variations of *CfD* and *CfE*. In this class *CfD* varied from −0.02 to 0.01 per year and *CfE* from −0.01 to 0.02 per year during the analyzed period.

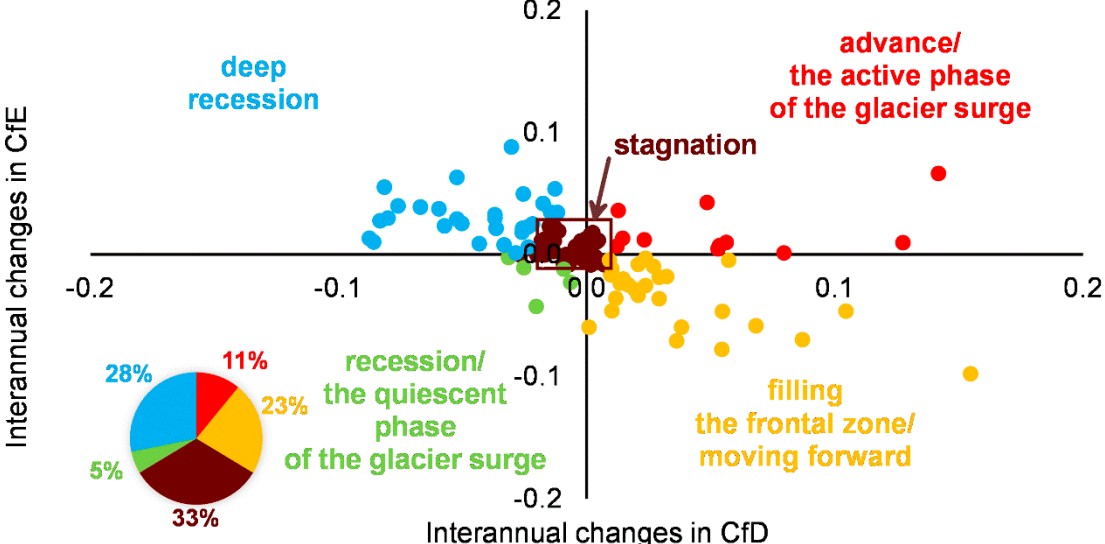

**Figure 8.** Classification of Spitsbergen tidewater glaciers according to the criterion of the glacier frontal zone dynamics, based on the state in the years 2014 and 2017—the relation of interannual changes of *CfD* and *CfE* and percentage representation (a pie chart inset).

A total of 28% of the 110 tidewater glaciers were in deep recession. In the 2014–2017 period, one-third were stagnant glaciers with an uncertain future behavior, although this might be a sign of an impending active phase of the glacier surge. For example, Nathorsbreen, Tunabreen, and Persejbreen showed interannual differences for *CfD* and *CfE* close to 0 about 4 years before the active phase of the glacier surge. Eleven percent of glaciers should have already been experiencing an active phase of the surge.

This classification was further compared with other information from these glaciers, particularly whether a clear advance was observed on Landsat satellite scenes in 2017 (6 glaciers), in 2018 (additional 7 glaciers), and in 2019 (additional 11 glaciers) (Figure 9). The advance was identified as a forward shift of the ice-cliff in relation to previous years, and based on the presence of a crevassed zone within the glacier frontal zone *Ag*, both characteristic of rapid glacier movement during the surge [7]. Eleven of the glaciers showed overlapping classifications as those clearly advancing and showing movement forward (Aavatsmarkbreen, Midtbreen, Vaigattbreen, Wahlenbergbreen, Arnesenbreen, Strongbreen → Moršneevbreen, Svalisbreen, Emmabreen, Tunabreen, Allfarvegen, and Moltkebreen), 7 as stagnant (Marstrandbreen, Nordenskiöldbreen, Kvalbreen, Sonklarbreen, Crollbreen, Markhambreen, and Recherchebreen), 5 as deeply withdrawn, ready for advance (Negribreen, Thomsonbreen, ve Osbornebreen, Fjortende Julibreen, and Lilliehöökbreen → Bjørlykkebreen), and 1—Monacobreen—as retreating. Five glaciers, classified as advancing, were still (2019) in recession/quiescent phase (Fjerdebreen, Chauveaubreen, the Heuglinbreen–Hayesbreen–Königsbergbreen system, and Nansenbreen) or stagnant (Kollerbreen). In the case of Johansenbreen, classified in Figures 8 and 9 as a stagnant glacier, an ice bulge is observed on satellite imagery (Landsat 8—2019) moving down glacier, while the medial moraine has been bent, both of which are typical first characteristics of glacier surge [19].

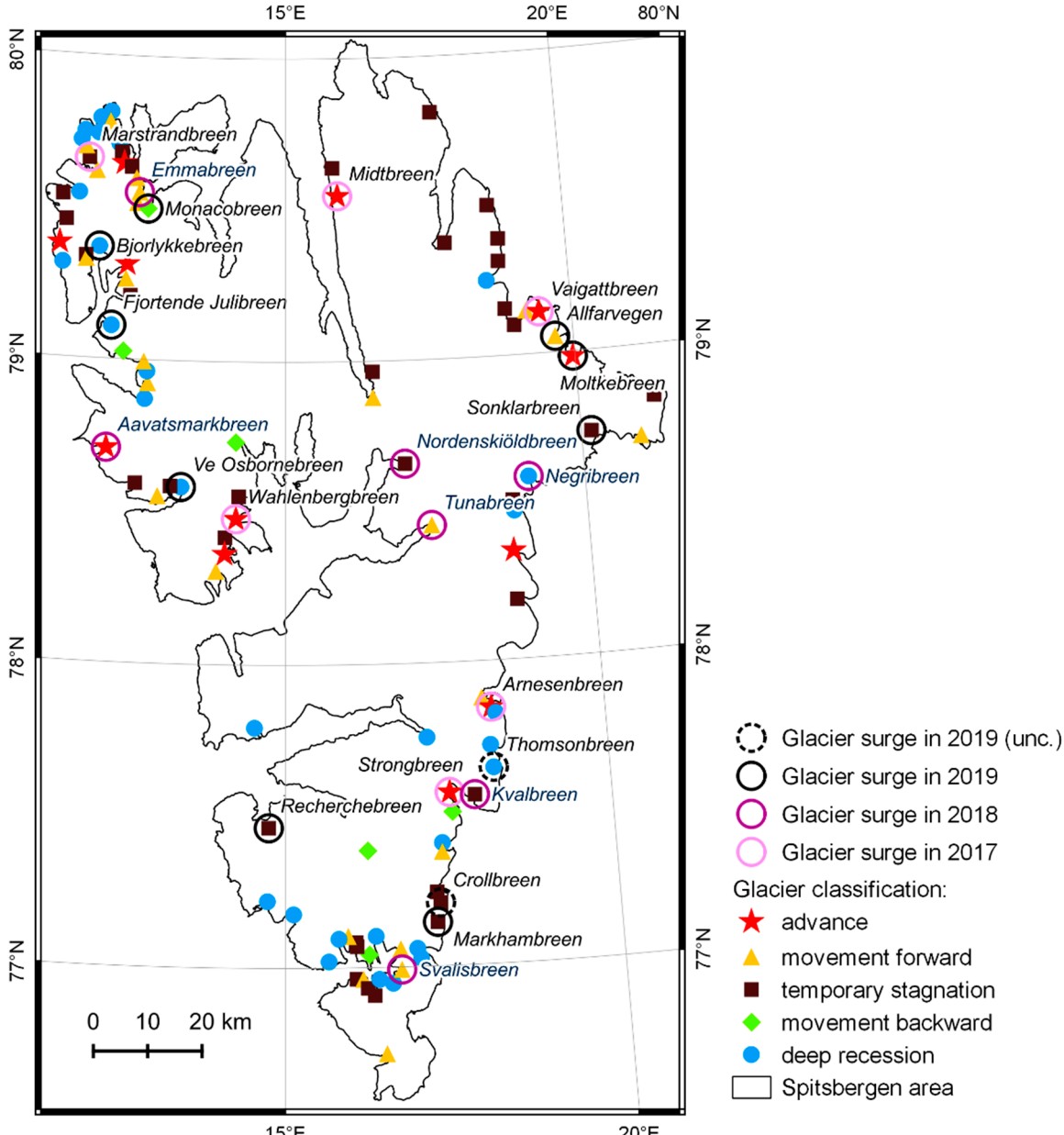

**Figure 9.** Classification of tidewater glaciers based on interannual changes of *CfD* and *CfE* between 2014 and 2017. Circles indicate glaciers with the characteristics of the active phase observed on satellite imagery in 2017, 2018, and 2019.

## 4. Discussion

### 4.1. The Role of the Ice-Cliffs Morphometry in the Glacier Surge Triggering

The recession of Spitsbergen tidewater glaciers is reflected in the morphometry of their ice-cliffs. The vast majority of them showed a retreating nature for the 1936–2017 period. This was reflected in the concave shape of the ice-cliff, which increases with the size of the glacier, as demonstrated by the *CfD* and *CfE* indicators. Approximately 10–18% of the tidewater glaciers have been experiencing the active phase at any given time, as manifested by the pronounced glacier front curving in the direction of flow (convex shape).

The most pressing question is at which critical point the forward glacier movement turns from normal flow to glacier surge. The preliminary analysis suggests that larger, complex glaciers (with many tributaries), with low average surface slopes and long ice-cliffs have a bigger tendency to surge. These

results also coincide with previous observations (e.g., [17,23,54–56]). The analysis also indicates that the bending of the ice-cliff will rapidly follow the initiation of the glacier surge in the case of large glaciers. However, this response tends to be delayed for smaller glaciers, up to five years, as the initial phase of the surge is characterized by a forward movement followed by a slow filling of the glacier frontal zone making a protrusion.

The frontal zone *Ag* of tidewater glaciers during the active phase usually fits into the *Ac* circle and the *CfD* indicator ranges from 0.5 to 1.0. The closer the ice-cliff is anchored at the valley mouth and the wider the valley, the more the *CfD* approaches 1. In extreme cases *CfD* even exceeds this value and the frontal zone "spills out" into the open sea (e.g., Negribreen in 1969). The median *CfD* in 1936 was the highest (cf. Table 1). At that time, tidewater glaciers had a greater range [4] and greater complexity of their glacial system. The high median *CfD* in 2000, in turn, was mainly the result of there being the largest number of cases of surging glaciers (18.2%).

The part of the glacier front that anchors the ice-cliff on land in an ice-wedge-like form can play an important role in the dynamics of tidewater glaciers. Their significance can be compared to buttresses like those found in Gothic cathedrals, designed to spread the force and support the weight. For glaciers, not normal forces (perpendicular to the glacier front line) are of particular importance for the stability of the strongly bending ice-cliff, but the bending moment and a cutting force follows a transverse to the glacier axis. Thicker and wider glacier buttress systems give the support and stabilization for the concave part of the glacier frontal zone (analogous to a parabolic arch, especially pointed arches in building structures [57]). Thus, larger glaciers also show a larger linear recession and greater ice-cliff bending, compared to smaller ones, before reaching the inflexion point (node) when, according to the catastrophe theory [58], the moment of the internal stresses is destabilized and the node (inflexion point) jumps. Therefore, larger glaciers are characterized by a rapid curving of the ice-cliff line following the glacier movement at the beginning of the active phase. Many glaciers, e.g., Nathorstbreen, Strongbreen, Storbreen, Mendelejeevbreen, Negribreen (cf. Figure 1), Selfströmbreen, Monacobreen (cf. Table 4), and Olsokbreen, showed a characteristic cliff indentation point(s) during deep recession, which could give rise to the stability node jump (e.g., by earthquakes, tides, waves or by the distribution of inner stresses etc.), initiating a hysteresis loop. The role of the "buttresses" in stabilizing ice-cliffs will increase as glaciers retreat into a shallower, sediment-covered valley bottoms, which may also contribute to extending the glacier surge cycle.

Smaller glaciers without buttresses only achieve a reduced front curving during recession, a so-called flat arch in terms of the ice-cliff shape. This arc is "structurally" less stable because it is more susceptible to normal forces [57]. Consequently, glacier dynamics appear to adapt faster to the interaction of the combined factors affecting the glacier–atmosphere–sea boundaries. The movement of the ice mass from the glacier top to the cliff [24,25] could be the prevailing force initiating the active phase in the case of smaller glaciers. This seems to be applicable to the approaching glacier surge of Johansenbreen, whose signs were seen on the Landsat 8 satellite image from 2019.

The model of glacier behavior before the glacier surge (mechanics of the structure) can be associated with the results of observations of the propagation of crevasses up the glacier from the ice-cliff [22,29,30]. The node jump of the ice-cliff line will also be favored by an asymmetry in the mass balance distribution, especially noticeable in meridionally located glaciers [59]. A disproportionate snow accumulation caused by prevailing winds and an uneven supply of direct radiation in summer (e.g., fog and shading) will also be reflected indirectly in the stress distribution throughout the entire glacier profile. Tidal amplitude [60] and fjord depth in front of the ice-cliff [37–39] also play important roles, especially in cases of anchoring the ice-cliff closer to the valley mouth. Thus, the greater the depth (hydrostatic support), the greater the impact on the cliff destabilization.

The glacier buttress system loses its function following surface ablation of land-based ice. The glacier surface altitude has been decreasing over the recent years, especially in the ablation area [61,62]. In turn, a thick layer of mineral deposits can work as an isolating layer protecting the ice beneath [63].

*4.2. Duration of the Active Phase During the Glacier Surge*

In the light of the observed Arctic amplification, the number of surging tidewater glaciers is expected to increase. Both, glacier coverage studies (e.g., [4–8]) and their *CfD* and *CfE* coefficients, indicate a deepening recession, including especially large glacial systems with the glacier buttresses and an extensive network of tributaries with a relatively narrow and flat receiving zone. One third of the glaciers are stagnant, and therefore potentially ready for an advance. The situation for 2017 morphometrically was similar to the situation in 1990 (cf. Table 1, Figure 4), which was followed by the culmination of cases of the active phase of the surge.

The beginning and end of the active phases of the glacier surge are the most difficult ones to determine through observation. This analysis shows that morphometric indicators can be a simple and accurate method to measure the duration of the active phase, as proven using published observational data. The active phase is estimated to last on average 3–10 years in Svalbard [14], based on published data, while the results presented here from 10 glaciers suggest an average duration of 6–10 years (see Table 2). The beginning and end of the active phase remains to be defined. A clear yearly jump in the value of the *CfD* and *CfE* coefficients and the positive sign in the *CfE* value are directly related to a sudden glacier forward movement, whose acceleration characterizes the active phase of the glacier surge. Therefore, this rapid increase and decrease in *CfD* and *CfE* coefficients (cf. Figures 6 and 7, Table 4) can be taken as a determinant of the beginning and end of the glacier surge.

The Monacobreen case study (cf. Table 4) shows that the proposed classification based on both indicators can be a useful tool for finding dynamic patterns of surging tidewater glaciers. These data refer to the results of Murray et al. [45], where the authors indicated three phases in the glacier surge cycle: acceleration, deceleration, and quiescence. Classification according to *CfD* and *CfE* indicates that both the active phase and the quiescent phase can show more variation in terms of the dynamic reaction of the glacier. The method requires further analysis to exclude overlapping of mid-seasonal fluctuations on interannual glacial dynamics. In this analysis, due to availability and quality, the data came from different parts of the ablation season.

## 5. Conclusions

The morphometric indicators used here, the frontal zone dynamics indicator *CfD* and the ice-cliff balance indicator *CfE*, allow determining of the dynamic state of tidewater glaciers and classifying them as: deeply receding glaciers, glaciers fulfilling the frontal zone/showing a forward movement, glaciers clearly advancing (active phase of the glacier surge), those showing signs of recession (quiescent phase), and stagnant glaciers. In addition, they allow forecasting future glacier behavior using remote sensing methods (e.g., satellite imagery, aerial photographs, laser scanning data, etc.).

The value of these indicators just before the surge is related to the size of the glacier and the complexity of the glacier system, the length of the ice-cliff and the average slope of its surface. The beginning and end of the active phase of the glacier surge can be identified as a sudden change in the ice-cliff morphometric indicators, as exceeding by several times the median of their interannual fluctuations in the order of approximately 0.05–0.06. This, in turn, gives the opportunity to estimate the duration of the active phase, which in the case of the 10 most-studied Spitsbergen tidewater glaciers appears to last on average 6–10 years.

The analysis shows a clear difference in morphometry and the behavior of the surging glaciers between smaller and large tidewater glaciers. One of the elements affecting the rate of the dynamic processes and their scale is the presence of glacier buttresses, which drives the distribution of the internal stresses in glaciers to the land anchored part, ensuring the stabilization of the ice-cliff. Buttresses are especially characteristic of large glaciers which can reach larger linear recession values, but also react more violently to the jump of the ice-cliff stability node. This also triggers the propagation of crevasses from the ice-cliff line up the glacier, especially in cases of anchoring the ice-cliff closer to the valley mouth. The presence of the glacier buttress system is probably another variable regulating

the distribution of forces inside the glacier, which should be included in the modelling of the calving processes of tidewater glaciers.

The morphometric model proposed here is of limited use for glaciers with very little contact with the sea and in the case of ice-cliff shading. Glaciers pinning on islands/capes are also a problem. In this case, each part of the ice-cliff between the island(s) and the land should be treated as a separate section. Glaciers that have not lost their complexity are best suited for long-term analyzes.

Currently, Spitsbergen tidewater glaciers show a strong recession pattern, as expected following Arctic amplification models. Thus, the morphometric indicators presented here also suggest a likely intensification of the phenomenon of the glacier surge in the coming years.

**Supplementary Materials:** The following are available online at http://www.mdpi.com/2076-3263/10/9/328/s1, Table S1: Emblems of topographic maps and IDs and acquisition dates of Landsat satellite images used in the research; Table S2: IDs and acquisition dates of Landsat satellite images for selected tidewater glaciers in Spitsbergen in the period 1985–2019.

**Funding:** This paper has been partly created thanks to funds of the Leading National Research Center (KNOW) No. 03/KNOW2/2014 received by the Centre for Polar Studies in the University of Silesia in Katowice for the period 2014–2018 and supported with in statutory activities No 3841/E-41/S/2019 of the Ministry of Science and Higher Education of Poland.

**Acknowledgments:** The author is very grateful to Leszek Kolondra (University of Silesia in Katowice) for sharing his collection of topographic maps. The author thanks very much the Reviewers and the Academic Editor, Kristian Kjellerup Kjeldsen, for all corrections and remarks improving the final version of the text.

**Conflicts of Interest:** The author declare no conflict of interest.

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
