# Peer review of "Ice-Cliff Morphometry in Identifying the Surge Phenomenon of Tidewater Glaciers (Spitsbergen, Svalbard)"

_geosciences, doi:10.3390/geosciences10090328_

Round 1

Reviewer 1 Report

Dear author and editor, 

I find the manuscript correctly written and presented, but I have some concerns about the methods and results, which I would like further clarification from.

Speaking about the coefficients, Cfd is an interesting one, although I miss its highest and lowest limits (I think that theoretically it does not have). Cfe, by contrast, does not seem interesting to me, first because it explains the same reality as Cfd, so it adds nothing valuable (hence the correlation in table 2). Second, its calculation, especially the moment it switches from positive to negative, is completely arbitrary (this is acknowledged in the text). As a result, when plotted in figure 6, there is a sudden leap on the graphs at surging events, but this is due to the mentioned arbitrary (and somehow misleading) change from negative to positive, and viceversa. 

In order to clear this suspicion, I would appreciate if the author could add a figure showing  the ice front during different years for any given glacier, as I believe that there is not a sudden shape change that justifies the large change in the Cfe value.

In addition, there are some parts in the manuscript that, to me, are unnecessary: 

  • Figure 7, and all its relevant comments in the text, are unrelated to the point of the paper, and show rather obvious information.
  • Likewise, point 4.1. is not related to the results of the research, but a summary of calving glacier dynamics which can be found in most Glaciology manual.   

I have also made some comments in the attached reviewed manuscript which I would also appreciate be answered.  

Reviewer 2 Report

The author has prepared a useful dataset of near-terminus geometric properties of tidewater glaciers. The use of CfD and CfE is well explained, and their use as proxies for advance and retreat are intuitive, although not proven here. It seems there should be a way of turning this data into a system for evaluating surging glaciers, and I encourage further exploration of these data. However, I think this paper is mostly about morphology without sufficient dynamical explanation.

The first sections and first two figures introduce the problem
clearly. Figure 3 is the first data-presenting graph. The early graphs (1936, 1961-1977) are difficult to compare with the
others because of using different sets of glaciers. Because the
clustering is similar in the last 4 times, one cannot help question whether we are seeing any time evolution or whether this is just characterizing different glacier geometries that do not vary much with time. What is needed is some analysis that ties the information in Figure 3 to the kind of information in Figure 6, in which we can show that glaciers move from one kind of CfE/CfD phase space to another position as they undergo surges. The data presented in Figure 6 show that this might be possible, but the analysis of medians in Figures 3 and 8 do not help.

Regarding some of the other figures: 4 does not show any significant trend in glacier morphology, 5 uses two many directional bins for the small number of glaciers, and 7b is a regression whose staistical power is almost entirely created by the presence of outliers. Also, do you really want to report 6 significant figures in Figure 7?

Despite my skepticism about the use of some of the figures, I think this paper contains potentially useful insights that are just not proven by the current presentation. In particular, if the sequence of events in section 3.4 can be fleshed out and shown in graphs based on the design of Figures 3, 6, and 8, then this will be a worthwhile contribution. As an example, see the well-known Madden-Julian Oscillation (MJO) graphs based on the work of Wheeler and Herndon (2004, see
https://www.cpc.ncep.noaa.gov/products/precip/CWlink/MJO/CLIVAR/clivar_wh.shtml for examples), then we could see how individual glaciers move within the phase space of Figure 8.

Finally, I dislike the use of enthalpy here, as it implies that the melt rate near the terminus is entirely based on local energy. If the glacier calves, then the ice can go elsewhere to melt, and the author discusses the fact that some of these fjords have significant advection of warmer water. Enthalpy seems misapplied here.

I am confident that this is a useful data set that needs to be turned into insights about the dynamics of the situation rather than just the morphology of the individual glaciers with no time connection between.

Round 2

Reviewer 1 Report

Dear author and editor: The paper has improved significantly with the removal of unnecessary parts and the addition of the Monacobreen example figure (Table 2). There are still two questions I would like be amended: - Now the Cfe calculation is clear, and I see it is not arbitrary. The issue about abnormal switch from strong positive to strong negative values remains, though, and it can be confirmed by Table 2. If Cfe was just an indicator, that would not be a problem at all, because it shows nicely when the ice front switches from concave to convex. However, you perform calculations and statistical analysis, as well as plot the values vs Cfd, and there it becomes problematic. If I understood properly, an ice front perfectly fitting the circle diameter will get a 1 value, an ice front very slightly concave will get a -0.9 value (for example) and a slightly convex ice cliff will get a 0.9 value. The values should range, in this case, from slightly negative for concave, through 0 for a straight ice front to slightly positive on convex. That way, the large yearly changes in the Cfd plots in figure 6, or the abnormal values (outliers) in the new figure 7 and in fig. 3, would probably not exist. Also, in my question about Figure 3: “How is it possible CFD values below 0?”, now I see the possibility of Cfd values below 0, but then the Ag explanation needs be improved because, as far as I could see, nowhere in the methodology it is said that Ag area out of the Ac circle accounts for negative values.
